# A First Case Report of Orbital Extra-Adrenal Paraganglioma in Cat

**DOI:** 10.3390/vetsci8050086

**Published:** 2021-05-14

**Authors:** Leonardo Leonardi, Raluca Ioana Rizac, Ilaria Pettinari, Luca Mechelli, Carlo De Feo

**Affiliations:** 1Department of Veterinary Medicine, University of Perugia, Via San Costanzo, 4-06126 Perugia, Italy; pettinariilaria@hotmail.it (I.P.); luca.mechelli@unipg.it (L.M.); 2Faculty of Veterinary Medicine, University of Agronomic Sciences and Veterinary Medicine of Bucharest, 105 Splaiul Independentei–District 5, 050097 Bucharest, Romania; ralucarizac@fmvb.ro; 3Veterinary Practitioner, Via della Moda, 4-06125 Perugia, Italy; cdf57@icloud.com

**Keywords:** paraganglioma, orbital, cat, extra-adrenal, cancer

## Abstract

Paraganglioma is a rare neuroendocrine neoplasm originating from paraganglia and consisting of neuroendocrine cells of the sympathetic and parasympathetic nervous system. Extra-adrenal paraganglioma occurs with a low incidence in both humans and animals. This report presents the first case of paraganglioma in a cat with orbital primary location. An 18-year-old spayed female European domestic shorthair cat of 3.60 kg body weight was evaluated in a private veterinary clinic in Perugia, Italy, for a pronounced exophthalmos of the right eye. The cat underwent surgery for the enucleation of the right eye and of the mass. The biopsy samples of the removed tissue were fixed in 10% buffered neutral formalin for histological and immunohistochemical evaluations. Therefore, specific markers were used for immunohistochemical investigations, such as anti-neuron specific enolase (NSE), anti-synaptophysin, anti-glial fibrillary acid protein, anti-cytokeratin and anti-chromogranin. The results of these investigations allowed establishing the final diagnosis of ocular extra-adrenal paraganglioma of the cat.

## 1. Introduction

Primary feline neuroendocrine and extra-adrenal paragangliomas are uncommon and rarely described orbital tumors. Paragangliomas (PGLs) are rare neuroendocrine tumors [1] composed of paraganglion cells associated with segmental or collateral ganglia arising from extra-adrenal paraganglia of the autonomic nervous system [2,3,4]. The paraganglion system is composed of a large network distributed symmetrically in the para-axial body regions within different tissues [5]. It is either grouped in cell aggregates, or in single individual cells, representing an important constituent of the neuroendocrine system [6], and it can be classified into four groups: the branchiomeric paraganglia (arteries and cranial nerves of head and cervical region), intravagal paraganglia—along the vagus nerve, the aortic-sympathetic paraganglia, visceral autonomic paraganglia (interatrial septum, liver hilum, duodenum, urinary bladder) [7].

Since PGLs develop in the autonomic nervous system, they originate in various parts of the body; for example, the most frequent ones are found in the carotid body and in the glomus jugular, while the less frequent ones in the orbit, vagus nerve, larynx, cauda equina, heart base, dorsal mediastinum and Zuckerkandl organ [8,9]. They can be classified as adrenal and extra-adrenal paragangliomas [8,10], and most of the paragangliomas are generated by the adrenal glands and only 10% are extra-adrenal. The latter derive from the parasympathetic ganglia [11], are usually more malignant than the adrenal ones [12], frequently occur in the head and neck regions [13,14] and are usually non-secretory [15]. Paragangliomas originate from the endocrine cells of the neural crest [2,16,17] and are characterized by the overproduction of catecholamines from adrenal and extra-adrenal tissues [18,19,20] which can cause, for example, palpitations, arrhythmias, weight loss, sweating and hypertension [21,22,23].

Clinically, nonspecific signs of orbital tumors are usually characterized by progressive unilateral exophthalmos, swelling and congestion of ocular tissues, frequent prolapse or protrusion of the nictitating membrane, conjunctival hyperemia, chemosis and keratitis. 

In addition, PGLs have a very low incidence in both humans and animals [10]. In fact, to date, they have been found and diagnosed mainly in dogs, while rarely in horses [6] and cats. Several cases of orbital PGL have been described in dogs and horses, including a case we also reported in 2019 in a quarter horse [6].

Establishing the diagnosis of PGLs requires special stains in addition to morphologic appearance on hematoxylin and eosin stained tissues. The neoplastic cells can be arranged into many different patterns, such as compact nests, trabecular, pseudo rosettes or pseudo glandular, and they are usually round to polygonal, clear to eosinophilic cytoplasm, stippled to hypercromatic nuclei [7].

The most frequent ways to diagnose PGLs are using special stains and immunohistochemistry (IHC). Argentaffin silver and argyrophilic stains might be used, since these tumors are argyrophilic and sometimes argentaffinic. If IHC is used, several antibodies must be chosen, since the tumor cells are positive for Synaptophysin, Chromogranin A + B, Neuron-specific enolase, Catecholamines and PGP9.5, and negative for Cytokeratin and Desmoplakins [7].

In this report, we describe a rare case of extra-adrenal orbital paraganglioma in a cat, diagnosed at the University of Perugia–Department of Veterinary Medicine.

## 2. Case Report

An 18-year-old spayed female European domestic shorthaired cat of 3.60 kg body weight was presented for a clinical evaluation due to a pronounced exophthalmos of the right eye. Clinical investigations and X-rays (digital radiological device TOP 30 HF 400 mA–99 kV) suspected the presence of a mass located in the retrobulbar space of the right eye. A transorbital ultrasound was proposed to confirm the diagnosis but was not accepted by the owner, as well as the proposed CT scan to better diagnose the mass and for a correct staging of it. The regional palpable lymph node was cytologically negative.

The first clinical examination on March 2019 revealed a normal sensory state, a mild grade of dehydration (3%), body temperature 38.2 °C, normal mucous membranes, refill time <2 s, cardiac rate 92 bpm, respiratory rate 38 breaths per minute, normal results for thoracic auscultation and abdominal palpation. The blood pressure measured by Doppler device was 180 mm Hg, but the cat was not cooperative. The hematological investigations revealed a lower reticulocyte hemoglobin value (11.8 pg) (Procyte Idexx), a hyperglycemic condition with a glucose value of about 196 mg/dL and an increased value of the glutamine pyruvic transaminase serum (ALT or SGPT)-175 U/L (Catalist Idexx).

The owners decided to subject the animal to surgery for the enucleation of the right eye and its subocular mass.

Transconjunctival enucleation of the ocular globe and removal of the retrobulbar mass were proposed, performing a lateral canthotomy and a 360° peritomy. An incision of the insertions of the extraocular muscles near the sclera was performed, until the retrobulbar mass was highlighted and the portion was clamped vascular. The orbital cavity was then filled with a hemostatic sponge, and the third eyelid and the conjunctiva were removed, then the eyelid margins were cut for a correct closure of the surgical wound (Figure 1). Sedation of the patient was performed with medetomidine and methadone IM, anesthetic induction with propofol IV, maintenance with isofluorane and fentanest in CRI, and upon awakening buprenorphine and meloxicam were administered with a cefazoin IV administration half an hour before induction. The patient awakened in a cage with a heated mattress and infrared lamp.

The second clinical examination, after the surgery, performed in May 2019, revealed an aggravated status of the cat, characterized by strong depression, anorexia, weight loss 3.4 Kg (2/5 BCS), dehydration, sialorrhea, vomiting, purulent discharge from the right nostril, widespread muscle pain, dysphagia, generalized tremors, ataxia. The patient was hospitalized and treated with saline solution IV, dexamethasone (0.5 mg/kg), lincomycin (11 mg/kg), maropitant (2 mg/kg), omeprazole (1 mg/kg), tramadol (2 mg/kg). The hematological investigations were characterized by erythrocytosis (12.63 M/μL), Red blood cells Distribution Width (RDW) (33.1%), low levels of eosinophils and platelets (0.07 and 92 KμL), glucose value of about 203 mg/dL. After 3 days, the patient was discharged due to the regression of clinical signs and a return to a normal clinical status, with a simple prescription of prednisolone and clindamycin preventive therapy.

## 3. Histopathology

The samples were fixed in 10% buffered formalin and sent to the diagnostic lab of the Department of Veterinary Pathology of the University of Perugia, also a reference center for the animal cancer registry of the Umbria region, for histopathological diagnostic investigations. 

Several fragments of tissue from the primary mass, brown in color and with a diameter varied from 0.4 cm to 1 cm, were collected. Subsequently, they were paraffin-embedded, 4–5 μm-sectioned using the microtome, and Hematoxylin-Eosin-stained. 

The histopathology investigations revealed a partially encapsulated infiltrative mass with multifocal angiotropic growth; the mass was composed of polygonal cells organized in nests, packs and bundles, all these structures being supported by a fine fibrovascular stroma (Figure 2). The lobules were lined peripherally by spindle cells. The tumor was highly vascular with blood-filled lacunae and multifocal to coalescing areas of liquefactive necrosis. The polygonal cells were characterized by indistinct borders, lightly eosinophilic cytoplasm with a moderate number of granules, round or oval central nuclei with stippled chromatin (“salt and pepper” appearance) and inconspicuous nucleoli. Anisocytosis and anisokaryosis were mild and mitoses were sporadic.

Following the histopathological findings, which had not allowed the issuance of a definitive morphological diagnosis, it was decided to continue a diagnostic process with further immunohistochemical investigations for a tumor referable to paraganglioma.

## 4. Immunohistochemistry

The tissue samples previously included in paraffin were cut into 2–3 μm sections. They were subsequently deparaffinized and rehydrated in distilled water. The antibodies used for IHC were: anti-Glial Fibrillary Acidic Protein (GFAP) (Dako, cod. Z0334, DakoCytomation, Denmark A/S, DK 2600, Glostrup, Denmark), anti-Neuron-Specific Enolase (NSE) (Dako, cod. M0873, clone BBS/NC/VI-HIG, DakoCytomation, Denmark A/S, DK 2600, Glostrup, Denmark), anti-Chromogranin A (Dako, cod. M0889, clone DAK-A3, DakoCytomation, Denmark A/S, DK 2600, Glostrup, Denmark) and anti-Sinaptophysin (Dako, cod. M0776, clone sy38, DakoCytomation, Denmark A/S, DK 2600, Glostrup, Denmark), anti-Vimentin (Dako, cod. M0725, clone V9, DakoCytomation, Denmark A/S, DK 2600, Glostrup, Denmark) and anti-Pancytokeratins (CK) (Cod. M3515, clones AE1/AE3, DakoCytomation, Denmark A/S, DK 2600, Glostrup, Denmark). All antibodies required pretreatment for antigen recovery except Chromogranin A. Therefore, the slides containing the histological sections were microwaved for 20 min at pH6 for anti-NSE and anti-GFAP antibodies, or at pH9 for anti-CK AE1/AE3 and anti-Sinaptophysin antibodies. Afterwards, the sections were treated with 3% H_2_O_2_ for inhibiting the activity of the endogenous peroxidase, and then incubated in a moist chamber with a protein block for 10 min. Incubation with primary antibodies was performed for two hours at room temperature, with the following dilution: anti-NSE (BBS/NC/VI-H14; Dako; 1:100, DakoCytomation, Denmark A/S, DK 2600, Glostrup, Denmark), anti-GFAP (Dako; 1:5000, DakoCytomation, Denmark A/S, DK 2600, Glostrup, Denmark), anti-CK (AE1/AE3; Dako; 1:200), anti-Synaptophysin (SY38; Dako; 1:30, DakoCytomation, Denmark A/S, DK 2600, Glostrup, Denmark) and anti-Chromogranin A (Dako; 1:200, DakoCytomation, Denmark A/S, DK 2600, Glostrup, Denmark). The sections were added with the secondary anti-mouse and anti-rabbit antibody of biotin, and finally streptavidin-peroxidase for 10 min. Antigen–antibody binding sites were identified by the chromogen aminoethyl carbazole (AEC). Finally, the sections were contrasted with Carazzi’s Hematoxylin and examined under a light microscope. Specific positive controls were used: brain cortex for NSE and GFAP, skin tissue for Cytokeratin, spinal cord for Synaptophysin and adrenal gland for Chromogranin A.

The tumor had diffuse strong cytoplasmic immunostaining to anti-chromogranin, anti-synaptophysin, anti-vimentin and anti-NSE, while anti-GFAP showed only multifocal small linear streaks among neoplastic cells, interpreted as cytoplasmic processes of sustentacular cells. The tumor was negative to anti-cytokeratin (Figure 3).

## 5. Discussion

In animals, PGLs are very rare neuroendocrine tumors and the cases described in the literature have been found in dogs [7,23,24] and horses [6,9,10,25,26,27] but only occasionally in cats or other animals. In fact, to the authors’ knowledge, only one case of paraganglioma of the cauda equina region of a cat has been described [3], but this neoplasm in the orbit of a cat has never been described.

In this report, the paraganglioma of the right orbit of an 18-year-old female cat has been described through clinical and pathological characteristics. 

The clinical signs exhibited by the cat were both general and local. The exophthalmos and the purulent discharge from the right nostril were the symptoms that were directly connected to the tumor. Other similar clinical signs that may occur are represented by decreased or absent vision, pale optic nerves, difficulty of eye retropulsion, chronic epiphora [26,27].

Histopathology revealed neuroendocrine cells with a nest organization as well as the presence of a fibrovascular stroma [28]. In addition, to make a correct and definitive diagnosis of this tumor, neoplastic cells must be positive for some immunohistochemical markers such as chromogranin A, NSE and Synaptophysin [4]. In fact, using immunohistochemical investigations, neoplastic cells are positive in different ways for the following biomarkers: Chromogranin A, NSE, SYN and GFAP. Chromogranin A, NSE and Synaptophysin are strongly expressed in neoplastic cells, while GFAP is poorly expressed. Nevertheless, it is useful for the final diagnosis since its expression decreases with tumor progression. 

From a diagnostic point of view, the paraganglioma always requires a series of supplements to the classical and morphological histopathological investigation, which demonstrates the need of immunohistochemical insights, as was also the case in our situation. From a macroscopic point of view, compared to the more “classic” forms described in the literature where these neuroendocrine tumors have been described as generally encapsulated benign forms, in our case we detected a mass of with multinodular and infiltrative growth characteristics that have represented a peculiarity of this form of tumor. From a histopathological point of view, on the other hand, no specific anomalies or particularities were detected compared to the standard descriptions. 

Several of the primary and metastatic orbital and retrobulbar tumors in cats are malignant and frequently associated with a poor prognosis. Distribution of primary and secondary orbital tumors in cats is described most frequently as secondary tumors by Attali-Soussay et al. [29]. Differential diagnosis must consider several other forms of neoplasia which are most frequently represented by tumors originating in the nasal cavities and sinuses, cheekbone-zygomatic tumors, frontal sinus tumors and primary tumors that most frequently affect felines, such as fibrosarcomas, osteosarcomas, lymphomas, and, above all, squamous cell carcinomas and poorly differentiated carcinomas. Obviously, the application of immunohistochemical investigation protocols with the use of the specific markers described represents the diagnostic method of excellence for the certain diagnosis of paraganglioma.

The immunohistochemical investigations performed were of fundamental importance, which detected a constant positivity to the markers against Synaptophysin, chromogranin, NSE and vimentin, while cytokeratin always gave a mild, sporadic marking. The diagnosis of certainty against a neuroendocrine tumor was reached thanks to the detection of the constant positivity of the neoplastic cells towards the three main markers used for neuroendocrine cells: Synaptofysin, Chromogranin and NSE, of which chromogranin is the endowed marker of higher specificity towards these cells, with the main expression at cytoplasmic granules level. This positivity may be milder in poorly differentiated tumors and with a higher degree of malignancy, but the positivity towards synaptophysin constant in all forms of neuroendocrine tumors with various degrees of malignancy, did not leave interpretative and diagnostic doubts in this case.

The neuroendocrine sustentacular cells also gave a multifocal positivity towards GFAP, testifying to the good differentiation of the paraganglioma in question, while the presence of the connective septa supporting the tumor allowed to detect a relative positivity also towards vimentin. Finally, the mild, focal, sporadic positivity for cytokeratin allowed us to exclude the presence of a form of neuroendocrine carcinoma.

The results highlighted by histopathological investigations are consistent and confirm a diagnosis of extra-adrenal paraganglioma of a cat’s orbit.

Less than two months after surgery for the removal of the retrobulbar tumor, the patient stopped eating and drinking and presented drooling, vomiting, severe dehydration, purulent discharge from the right nostril and after a few days a serious nervous symptomatology arose: loss of coordination and inability to avoid obstacles, evidenced by the loss of the sign of threat. Due to the serious deterioration of the animal, which reached a precomatose state, euthanasia was decided by the owner. The body was not available for a necropsy examination. 

Due to the vast distribution of the paraganglia and the numerous histopathological patterns that the PGLs might embrace, the diagnosis for this type of tumor must be established using the “gold” IHC kit.

## Figures and Tables

**Figure 1 vetsci-08-00086-f001:**
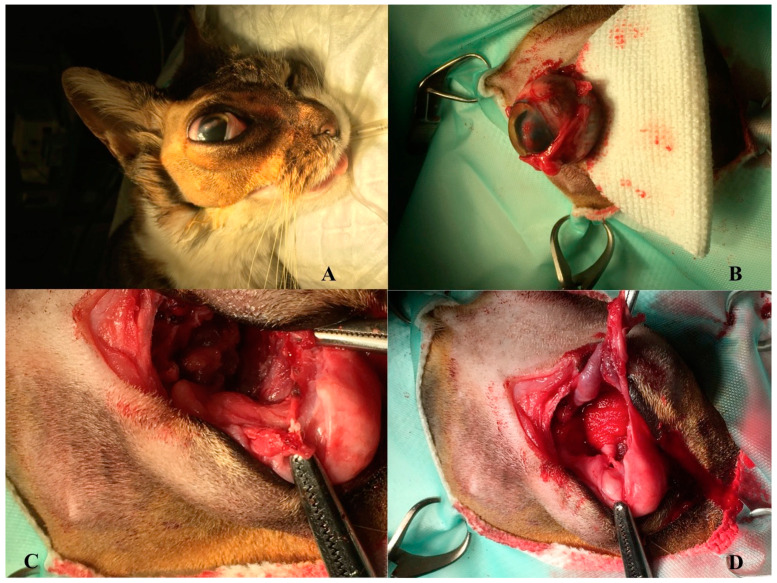
Gross evidence of the mass (exophthalmos) before surgery (**A**). Enucleation of the right ocular globe during the surgery (**B**). Gross evidences of the primitive multilobular mass before the total surgical excision (**C**,**D**).

**Figure 2 vetsci-08-00086-f002:**
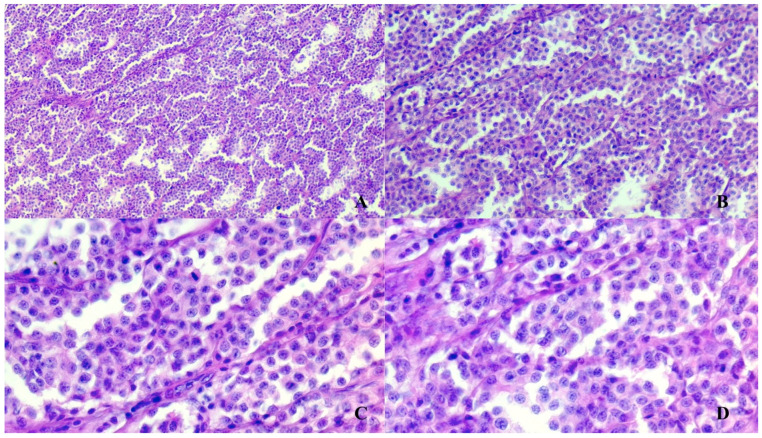
Paraganglioma histopathological features. Typical structural pattern of paraganglioma organized in packs, nests and bundles of spindle cells supported by fibrovascular stroma. Hematoxylin-Eosin stain, magnification of 4× (**A**), 10× (**B**), 20× (**C**,**D**).

**Figure 3 vetsci-08-00086-f003:**
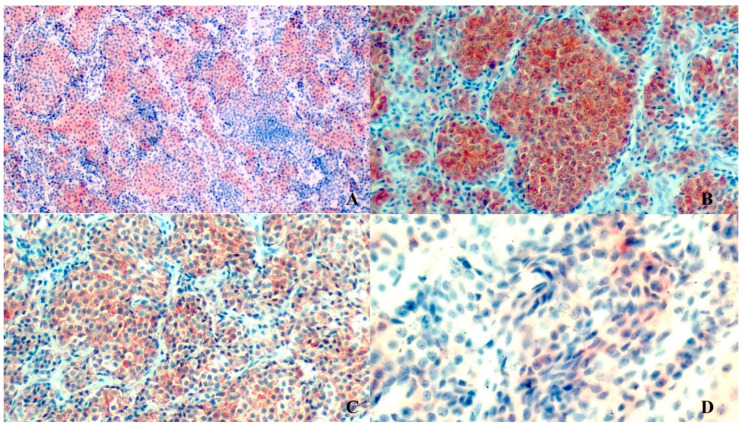
Paraganglioma features for immunohistochemical markers. (**A**) Anti-synaptophysin. (**B**) Anti-Chromogranin. Anti-NSE. (**C**) Anti-NSE (**D**) Anti-Cytocheratins. (**A**) (magnification 10×), (**B**,**C**) (magnification 20×) figures show a diffuse cytoplasmic immunostaining marking of neoplastic cells of paraganglioma. (**D**) (40× magnification) anti-cytokeratin stain shows a very mild and focal positivity (negative) stain. Immunoperoxidase.

## Data Availability

The data presented in this study are available in the manuscript.

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
