# Peer review of "A First Case Report of Orbital Extra-Adrenal Paraganglioma in Cat"

_vetsci, 2021, doi:10.3390/vetsci8050086_

Round 1
Reviewer 1 Report
The manuscript presented a case report of orbital extra-adrenal paraganglioma in a cat. Apparently, this report would be the first description of this neoplasm in a cat, which is interesting and would support the possibility of publication. However, the work is presented in a very confusing way. The introduction section should be redone. This section should define what a paraganglioma is and its most frequent locations. In veterinary medicine, the orbital paraganglioma is well documented in horses, nevertheless, this information is not mentioned.The description of the surgery is poor, however, there are three pictures of this procedure, which added nothing to the work. Most histopathology and immunohistochemistry images are out of focus (blurry) and should be replaced. Figure 2 has no caption. In my opinion, the picture of immunohistochemistry for synaptophysin does not seem convincing, showing real immunoreactivity, but rather a nonspecific reaction.In most of the discussion section, the authors only present the information but do not discuss it. It would be interesting to discuss the main differential diagnoses for tumor-like lesions in the cat's orbits. In the discussion section, the authors report what happened to the cat after surgery. This information must be contained in the results. Still in this part, the authors raise the hypothesis of a possible cerebral metastasis of this tumor, however this is totally speculative information. Thus, I do not recommend this article for publication in Veterinary Sciences.
Author Response
Dear Reviewer 1, first of all thank you very much for your precious and kind suggestions. We have tryed to correct everything you underline on hope to improve our work. We pedone introduction, we improve description of surgery, we change some pictures with new ones, as like as synaptophysin and not focusing ones. we inserti also dome data related to differential diagnosis as like as you advice us. We hope this new version will be suitable for your kind positive evaluation. Thanks again.

Reviewer 2 Report
Dear Author,
please correct the title, in my opinion, is illogical, either rare case or first case, please choose. Photos in Figure 2 are out of focus, especially D. No magnification given, no scale. In figure 3 please change to larger magnification. Please provide a magnification. In photo A the reaction is questionable, please correct.
In my opinion introduction, results and discussion are well written. All comments and remarks are in the attached file.
Regards
Author Response
Dear Reviewer 2, thank you very much for your precious suggestions. We correct as you advice the title and above all the picutures (Apologize us for the misunderstanding about it), we add magnifications and more details. Thanks again.

Reviewer 3 Report
General: The paper by Leonardi and colleagues is a succinct description of a rare neoplasm (extra-adrenal paraganglioma) in a domestic shorthair cat. It purports to be the first publication of this neoplasm in the orbital region in a cat. The case report needs significant editing for English syntax and grammar as well as spelling and the authors are encouraged to seek the help of an English-speaking colleague. The case report would be significantly enhanced by inclusion of some additional clinical information and specifics in addition to pure pathology description. Details and specificity are needed to bolster the clinical description.
Specific:
Title line 2 - No need to put “Rare” in title as this is implied by the fact that this is the first reported case in the literature.
Abstract: Line 15 - remove name of veterinary clinic as not pertinent
Abstract: Line 16 – remove sentence starting “The owner was advised…….” – not meaningful to case
Introduction line 50 – Change “Histopathology is an intermediate tool” to Definitive diagnosis of PGLs requires special stains in addition to morphologic appearance on hematoxylin and eosin stained tissue.
Introduction line 53 – pathology terminology needs to be proper (part of the English revisions needed in the paper) “hipercromatic nuclei” should be revised to hyperchromatic nuclei.
Introduction line 56 – Remove “a bunch of antibodies” and re-word
Case report – lines 62-68 – Add specificity. Terms like “normal general clinical parameters” mean little. What clinical parameters were evaluated? If imaging such as radiology was utilized than what was examined? Orbital region? Brain? etc.. This is a PGL so was blood pressure evaluated? Cardiac parameters? What is normal for one clinic may not be normal for another. If a description is used like weight loss than how much? In the discussion section on line 198 of discussion states “serious nervous symptomology arose” – specifically what were the neural signs?
Equipment and methods used should be explicitly stated (eg. clinical pathology tests and instrumentation), doses and sources of drugs etc. etc.
Figure 1 – photos such as B showing enucleated globe and D final wound closure not really useful
Figure 2 – needs figure legend. Photomicrographs C &D are out of focus. Only really need B and D to show a low magnification of the tissue architecture and a high magnification of the cytologic appearance. Should have magnification bars on photomicrographs.
Figure 3 – the photomicrographs need legend and should have magnification presented as above.
Immunohistochemistry section – Were sections really made at 2-3 microns for IHC yet 4-5 microns for H&E? Unlikely to perform microtomy on a paraffin block at 2 microns. A table should be provided showing exact antibodies (sources, clones etc.) used and some reference of how it was determined that they were appropriate for feline tissue such as references? Conditions of fixation (how long in 10% neutral buffered formalin?) should be mentioned as well somewhere in the case report.
The literature is not complete as there are additional reports of extra-adrenal PGLs in cats. Should be included.
Author Response
Dear Reviewer 3, thank you very much for your precious help and collaboration. We tried to improve everything you suggest with a generale revision, several add especially in the Clinical description of the case and tried to correct point by point of Specific topics you sent us. Apologize us for the problem related to pictures but we didn't understand what happened when we sent the original paper. Now we change also the pictures out of focus with new ones. Again thank you. I ensure to Editor to send back the paper today but I'm out of Perugia and I cannot check about (sources and clone of antibodies). The last data that I still miss are these, please be patient.

Round 2
Reviewer 1 Report
I appreciate the changes and improvements done by the authors. All previous comments have been addressed appropriately and information was added accordingly.
Author Response
Dear Reviewer 2, we are very glad to look about your appreciation and we would like to thank you very much again for you rprecious help and suggestions to improve the paper.
Best regards.

Reviewer 3 Report
Needs editing for English syntax and grammer
Author Response
Dear Reviewer 2, we are very glad to receive your comments and we would like to thank you very much again for your precious help and suggestions to improve the paper. We try to edit again the syntax and grammar as you suggested.
Best regards.
